

# Characterisation of microsatellite and SNP markers from Miseq and genotyping-by-sequencing data among parapatric *Urophora cardui* (Tephritidae) populations

Jes Johannesen[1], Armin G. Fabritzek[1], Bettina Ebner[2] and Sven-Ernö Bikar[2]

[1] Institute of Organismic and Molecular Evolution, Mainz University, Mainz, Germany
[2] StarSEQ GmbH, Mainz, Germany

## ABSTRACT

Phylogeographic analyses of the gall fly *Urophora cardui* have in earlier studies based on allozymes and mtDNA identified small-scale, parapatrically diverged populations within an expanding Western Palearctic population. However, the low polymorphism of these markers prohibited an accurate delimitation of the evolutionary origin of the parapatric divergence. *Urophora cardui* from the Western Palearctic have been introduced into Canada as biological control agents of the host plant *Cirsium arvense*. Here, we characterise 12 microsatellite loci with hexa-, penta- and tetra-nucleotide repeat motifs and report a genotyping-by-sequencing SNP protocol. We test the markers for genetic variation among three parapatric *U. cardui* populations. Microsatellite variability ($N = 59$ individuals) was high: expected heterozygosity/locus/population (0.60–0.90), allele number/locus/population (5–21). One locus was alternatively sex-linked in males or females. Cross-species amplification in the sister species *U. stylata* was successful or partially successful for seven loci. For genotyping-by-sequencing ($N = 18$ individuals), different DNA extraction methods did not affect data quality. Depending on sequence sorting criteria, 1,177–2,347 unlinked SNPs and 1,750–4,469 parsimony informative sites were found in 3,514–5,767 loci recovered after paralog filtering. Both marker systems quantified the same population partitions with high probabilities. Many and highly differentiated loci in both marker systems indicate genome-wide diversification and genetically distinct populations.

## INTRODUCTION

*In situ* parapatric divergence across an environmental gradient is expected to result in a combination of non-concordant genetic clines among loci under selection (and loci linked to them), due to different selection coefficients on these loci, and a majority of "neutral" loci structured by a balance between genetic drift and gene flow (*Coyne & Orr, 2004*). Because environmental adaptation is expected to act on few loci only, genetic divergence across

Corresponding author
Jes Johannesen, jesjo@uni-mainz.de

the genome should create "islands" of divergence rather than genome-wide divergence patterns in the presence of gene flow. However, recent genomic studies of incipient species with gene flow suggest the latter (*Michel et al., 2010*; *Renuat et al., 2013*), in contradiction to expectations.

*Urophora cardui* L. (Diptera: Tephritidae) is a specialist gall maker associated with creeping thistle *Cirsium arvense* (L.) Scop (Asteraceae). Initially attracting attention as a biological control agent of *C. arvense* in Canada (*Zwölfer, Englert & Pattullo, 1970*; *Peschken & Harris, 1975*), *U. cardui* has since been studied for a range of topics in its native Western Palearctic and introduced Canadian distribution ranges: biogeography, evolution of galls, interactions with parasitoids and for pest control (e.g., *Peschken & Harris, 1975*; *Zwölfer, 1979*; *Eber & Brandl, 1997*; *Johannesen, Drüeke & Seitz, 2010*; *De Clerck-Floate & Cárcamo, 2011*).

The local population structure of *U. cardui* in the native range is characterised by meta-population dynamics and isolation-by-distance gene flow (*Zwölfer, 1979*; *Seitz & Komma, 1984*; *Eber & Brandl, 1994*; *Eber & Brandl, 1996*; *Johannesen & Seitz, 2003*). However, *Steinmetz, Johannesen & Seitz (2004)* reported a narrow (ca. 70 km), linear genetic transition zone across contiguously distributed populations on the Jutland (Cimbrian) peninsula where genetic variance between populations on each side of the transition highly exceeded that explained by recurrent extinction-colonisation processes. The study, based on allozyme loci, identified non-concordant clines at three loci and no clines at another four. *Johannesen, Drüeke & Seitz (2010)* reported dominance of the same mtDNA haplotype on both sides of the transition zone that belonged to a Western European lineage experiencing population expansion. This mix of allelic distributions across the transition area suggested an *in situ* origin of the parapatric divergence rather than one caused by secondary contact of two allopatrically evolved populations, i.e., by vicariance. However, a precise estimate of the relative number of loci exhibiting clines as well as modes of allele frequency shifts was prohibited due to low levels of polymorphism.

Here, we report the development of 12 highly polymorphic microsatellite loci and a genotyping-by-sequencing (GBS) protocol for SNP identification with the aim of reaching first insights into genome-wide vs. island diversification among parapatric *U. cardui* populations, specifically, and for the study of population genetic patterns in native and introduced populations of *U. cardui* in general. The two marker systems are characterised by different levels of per-locus polymorphism and they differ in locus-specific recovery (amplification) per individual. Microsatellites have great applicability due to locus-specific amplification per individual, high allelic variation per locus and Mendelian inheritance. In comparison, GBS provides high-density, genome-wide (mostly) bi-allelic polymorphisms, but the recovery of specific loci (locus amplification success) may be uneven among individuals. The markers may, alone or in combination, offer helpful tools relative to the specific scientific enquiry and/or sampling strategy (*Hodel et al., 2016*). We quantify genetic diversity and differentiation among one population north, within and south of the transition zone, respectively, and we assess the information value of the markers systems relative to allozyme genetic divergence observed 14 years previously (*Steinmetz, Johannesen & Seitz, 2004*).

## MATERIAL AND METHODS

### Microsatellites

Microsatellite development was based on an Illumina-Miseq DNA library generated from one male caught in the centre of the transition zone, Østre Løgum (55.133°N, 9.351°E). Genomic DNA was extracted with the Roche template preparation kit (Roche Diagnostics, Mannheim, Germany). 100 ng of genomic DNA was sheared, end repaired, A-tailed and ligated to TruSeq adapters. The library was amplified in 8 PCR cycles. The library was size selected for a mean of 650 bp, which corresponds to 530 bp internal sequence. The library was sequenced for 300 bp in "paired-end" modul in one Illumina-Miseq run. In total, 53.5 million paired-end reads (16.2 GB) were generated (NCBI BioProject: PRJNA352663; SRA: SRR5023787). Overlapping paired end reads were merged with FLASH v. 1.2.6 (*Magoc & Salzberg, 2011*), and the sequences were searched for microsatellite repeat motifs. Initially, 117 loci with repeat motifs were identified. PCR amplification for the 117 Loci was performed in 25 $\mu$l: 1 $\mu$l DNA (15 ng), 1 $\mu$l forward and 1 $\mu$l reverse primer (10 pmol/$\mu$l), 2 $\mu$l dNTPs (2.5 mM), 5 $\mu$l 5× PR-Buffer, 2 $\mu$l MgCl (25 mM), 0.1 $\mu$l GoTaq-Hotstart-polymerase (1.25 units) (Promega, Fitchburg, WI, USA), 12.9 $\mu$l $H_2O$ using Biometra thermocycler T1 (Biometra, Göttingen, Germany): 3 min at 95 °C, 34 cycles of 30 s at 95 °C, 30 s at 55 °C, 30 sec at 72 °C, final extension of 5 min at 72 °C. 53 loci produced distinct PCR products. Variability in the 53 loci was assayed by comparing PCR products from a mix of four individuals with the PCR product of the original individual. The PCR products were separated electrophoretically using the QIAxcel-System (Qiagen, Venlo, The Netherlands). Twelve loci with hexa-, penta- and tetra-nucleotide repeats were optimised for analyses (Genbank accession numbers KT923909–KT923920) (Table 1). Alleles with such repeat units are easier to score (bin) accurately than alleles with dinucleotide repeats. The 12 loci were amplified in three QIAGEN Multiplex PCR reactions with fluorescent-labelled primer pairs (primer mix 1: Uc03, Uc05, Uc09, Uc10; primer mix 2: Uc01, Uc02, Uc08, Uc11; mix 3: Uc04, Uc06, Uc07, Uc12) (Table 1). For multiplex PCR, we used an end volume of 10 $\mu$l (8.5 $\mu$l mastermix and 1.5 $\mu$l DNA of *c.* 50 ng DNA per reaction). The mastermix contained a final concentration of 1× QIAGEN Multiplex PCR Master Mix, which had 3 mM $MgCl_2$ and 0.2 $\mu$M of each primer. Cycling conditions were identical for the three multiplex reactions: 30 s at 95 °C, 30 cycles of 30 s at 94 °C, 90 s at 60 °C, 90 s at 72 °C, final extension of 10 min at 72 °C. 1 $\mu$l of the PCR product was added 11.7Rl HiDi-formamide and 0.3Rl ROX 500 standard (Applied Biosystems, Carlsbad, CA, USA). The loci were run on an ABI 3130XL capillary sequencer and genotyped with GeneMapper 5.0 (Applied Biosystems, Foster City, CA, USA).

### GBS

DNA was digested with the restriction enzyme EcoR1. Based on *in silico* digestion of c. 1 Mb of the tephritid *Ceratitis capitata* (GenBank assembly accession: GCA_000347755.2) and *Bactrocera tryoni* (GCA_000695345.1) genomes, we estimated that EcoR1 would have c. 15,000 fragments between 200 and 500 bp. DNA for the GBS library was extracted with the Roche template preparation kit (Roche Diagnostics, Mannheim, Germany). DNA quality and its interaction with GBS library quality (sequencing success) was assessed for extractions

**Table 1** Microsatellite loci developed for *Urophora cardui*.

| Locus | Genbank accession no. | Primer sequence 5′–3′ | Tm (°C) | Core repeat | Size range (bp) | PM |
|---|---|---|---|---|---|---|
| Uc01 | KT923909 | F: NED-TTAAGCATTAACGGACCAGAAG | 57 | (TTGTTT)9 | 293–326 | 2 |
| | | R: CTAGGAGTGGCTATGCGGG | 60 | | | |
| Uc02 | KT923910 | F: HEX-ATGTCGATTACACTGTGCTTC | 57 | (GAAAA)16 | 308–420 | 2 |
| | | R: TGCTCACTTCTGGTGGC | 57 | | | |
| Uc03 | KT923911 | F: FAM-TCGCACTTCTGGGATGGAG | 60 | (TTTATG)9 | 247–342 | 1 |
| | | R: AGGCAATAGTCTTATGCACAGC | 60 | | | |
| Uc04 | KT923912 | F: HEX-GGACCTATTAGATGGAGCTGG | 58 | (GTAAAG)11 | 301–358 | 3 |
| | | R: TCGGACCATAATCACGCCC | 60 | | | |
| Uc05 | KT923913 | F: HEX-TGTACTGTGCTACACGCGG | 60 | (AAATG)10 | 143–178 | 1 |
| | | R: CACTTGCATCTGCCAGCC | 60 | | | |
| Uc06 | KT923914 | F: FAM-GGCCTTGATCAGGACTTCAAC | 60 | (TAACT)15 | 259–319 | 3 |
| | | R: AACGCGTGTGTATCGAGGC | 61 | | | |
| Uc07 | KT923915 | F: NED-TCGATGCTTTCCTTCTGTCAAC | 60 | (CTTCA)10 | 376-446 | 3 |
| | | R: GTGCAGCTCAAGTGCTAATAAAC | 59 | | | |
| Uc08 | KT923916 | F: FAM-AATTGGCGCCTTTCTGCAC | 60 | (TAAGAA)11 | 382–550 | 2 |
| | | R: GCACAGTGGGACGAAACTC | 60 | | | |
| Uc09 | KT923917 | F: NED-AGCAAACATTCTCTGAGCCC | 59 | (CAATA)12 | 276–316 | 1 |
| | | R: TCGGATTCGATCCAGGCAC | 60 | | | |
| Uc10 | KT923918 | F: HEX-CTTGTCAGCCTGTGCATACC | 60 | (CTATT)17 | 301–431 | 1 |
| | | R: ACGAAGCGTGCCATTCAAG | 60 | | | |
| Uc11 | KT923919 | F: FAM-GACAAGCATGTTGCTAAAGCG | 59 | (ACTCAT)11 | 224–320 | 2 |
| | | R: AGCCATCTGCATTTGTTGG | 57 | | | |
| Uc12 | KT923920 | F: NED-CGACGATATGTATGTACCAGAGG | 59 | (GATA)17 | 198–262 | 3 |
| | | R: AATGAGCTGGAGGGCACAG | 60 | | | |

**Notes.**

Tm, melting temperature; PM, primer mix.

of (1) whole individuals cut longitudinally and treated with RNAase (Qiagen, Venlo) after DNA extraction, and (2) for DNA extracted from ultrasonically lysed (Bandelin Sonoplus HD70) tissue pellets without RNAase treatment. GBS adapter preparation followed *Elshire et al. (2011)*. We used three pools with six barcodes (individuals) each (Appendix S1). Individual DNA was digested separately in 20 μl end volume consisting of 2 μl NEB buffer (10×), 1 μl EcoR1 (NEB) and 100 ng/μl DNA (variable μl), and filled up to 20 μl with RNase free water (variable μl). The mixture was gently stirred by flicking, centrifuged for 30 s (8,000 rpm), incubated at 37 °C for 15 min and heat inactivated at 65 °C for 20 min. Ligation was done for each individual separately in 50 μl end volume consisting of 20 μl digest (above), 5 μl NEB ligase buffer (10×), 10 μl barcoded adapter mix (=concentrated stock), 13.5 μl RNase free water, 1.5 μl NEB T4 DNA ligase. Reactions were centrifuged for 30 s (8,000 rpm) and incubated at 22 °C for 60min. The ligase was heat inactivated at 65 °C for 30 min. The ligation products of six individuals (each 10 μl) with different barcodes were pooled (=60 μl), added 250 μl binding buffer (Roche) and eluted in 30 μl RNase free water. Each pool was amplified in 6 parallel PCRs (with one specific TruSeq primer per pool). PCR purification was done on three PCR reactions simultaneously using 3 × 250 μl = 750

µl binding buffer, and eluted in 30 µl RNase free water. Finally, the two 3× purifications á 30 µl were pooled, resulting in 60 µl end-elution product. DNA concentrations were measured with Qubit 3.0 (Thermo Fisher Scientific, Waltham, MA, USA), DNA quality was assessed with the 260/280 ratio using Nanodrop (PEQLAB, Erlangen, Germany). Library validation was done with the Agilent 2,100 bioanalyzer (Santa Clara, CA, USA). Libraries were sequenced as 100 nt paired ends on Illumina Hiseq 2000 (1/5 lane, three pools on six individuals) at the Institute of Molecular Genetics, Mainz University.

GBS pools were de-multiplexed with pyRAD 3.0.3 (*Eaton, 2014*). The paired end reads were merged with PEAR 0.9.5 (*Zhang et al., 2014*) before being processed further with pyRAD 3.0.3, using standard settings for paired end reads. Data processing was done on the High Performance Computing (HPC) Mogon-cluster (Mainz University). We assessed the number of recovered loci, unlinked SNPs and parsimony informative sites by varying combinations of the parameters "minimum coverage of individuals" (mincov = 10 and 14), and "maximum number of individuals with heterozygotic sites" (maxSH = 4, 6, 8 and 10).

### Genetic analyses

Genetic diversity indices were estimated in three populations situated north (Vildbjerg, 56.187°N, 8.771°E), within (Frøslev, 54.827°N, 9.331°E) and south (Neumünster 54.136°N, 9.907°E) of the transition zone. Microsatellite loci were tested in 19–20 individuals per population. GBS SNP variation was quantified for 6 individuals per population. Cross-species amplification of microsatellite loci was tested in *U. stylata* (Fabricius 1775) that were sampled in Christiansfeld, Denmark (55.354°N, 9.476°E) and Waldgrehweiler, Germany (49.664°N, 7.737°E). For microsatellites, we tested for deviations from Hardy–Weinberg proportions (per locus and population), linkage disequilibria (LD) within populations, genetic diversity per population (allele number and expected heterozygosity) with Genepop on the web (*Raymond & Rousset, 1995*; *Rousset, 2008*). Genetic differentiation, $F_{ST}$, was estimated locus-wise and among populations with Arlequin 3.5.2.2 (*Excoffier & Lischer, 2010*). Microsatellite null alleles, stuttering and large allele dropout were evaluated with Micro-Checker version 2.2.3 (*Van Oosterhout et al., 2004*). For GBS data, we converted the Structure file of unlinked SNPs generated by pyRAD (*Eaton, 2014*) into an Arlequin file (PGDspider 2.1.0.1, *Lischer & Excoffier, 2012*). For both microsatellites and GBS, we assessed population affiliations with Structure 2.3.4 (*Pritchard, Stephens & Donnelly, 2000*) using the admixture model, both with and without prior population information (10 × 50,000 burn-in and 100,000 iterations, $K = 1$–5). The highest level of genetic clustering was calculated in Clumpak (*Kopelman et al., 2015*) according to delta $K$ (*Evanno, Regnaut & Goudet, 2005*) and probability of $K$, ln Pr$(X|K)$ (*Falush, Stephens & Pritchard, 2003*).

## RESULTS

### Microsatellites

All microsatellite loci amplified consistently in the 59 investigated *U. cardui*. The tetra, penta and hexanucleotide repeat lengths allowed precise binning of alleles. Both standard

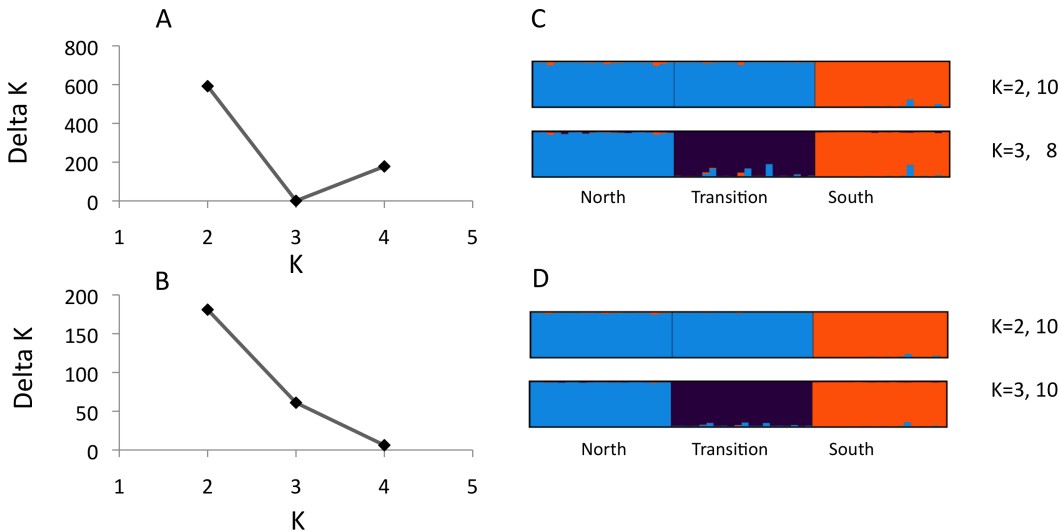

**Figure 1 Structure analysis of microsatellite data for three parapatric populations (North $N = 20$, Transition $N = 20$, South $N = 19$)** Optimal number of $K$ clusters estimated using the admixture model, without (A) or with (B) prior population information (*Evanno, Regnaut & Goudet, 2005*). Individual memberships for $K = 2$–$3$ without (C) and with (D) prior population information are visualised by different colours. Numbers after $K$ (e.g., 10) represent the number of partitions found in 10 runs. Calculations were done with Clumpak (*Kopelman et al., 2015*).

fragment length variation as well as intra-repeat variation, e.g., 3 bp shifts in hexanucleotide repeats, were easily discriminated. Because all loci were amplified with identical PCR conditions, other multiplex locus combinations could be possible in future studies.

We observed 5–21 alleles per locus across the three *U. cardui* populations (Table 2). The expected heterozygosity per locus per population was 0.60–0.90. Locus Uc12 was differentially sex-linked: both sexes lacked heterozygotes in Vildbjerg north of the transition zone ($F_{IS} = 0.874$), predominately in females in Frøslev within the transition zone ($F_{IS} = 0.389$) but in males in NMS south the transition zone ($F_{IS} = 0.418$) (Table 2). $F_{IS}$ values of 0.50 are expected when haploid and diploid loci (or organisms) are equally represented in analysis. Loci Uc05 and Uc06 did not obey Hardy–Weinberg proportions in the population Frøslev after Bonferroni correction ($P < 0.05$). Linkage disequilibria were not found in the northern population but they were observed in the transition zone (Frøslev, $N$ pairs $= 6$) and in the southern population (NMS, $N$ pairs $= 1$) after Bonferroni correction ($P < 0.05$). Most LD involved the sex-linked locus Uc12. There was no evidence for stuttering or large allele drop out at any locus. The method of *Evanno, Regnaut & Goudet (2005)* identified $K = 2$ as the highest level of structure (Figs. 1A and 1B) whereas Clumpak summary results revealed an approximately equal probability for $K = 2$ and $K = 3$ (Figs. 1C and 1D), particularly when including population priors. For $K = 2$: mean(LnProb) $= -2855.010$, mean(similarity score) $= 0.997$; for $K = 3$: mean(LnProb) $= -2694.350$, mean(similarity score) $= 0.989$. Mean expected heterozygosity/population across loci was North $H_e = 0.788$, Frøslev $H_e = 0.810$, South $H_e = 0.792$. All 12 loci were significantly differentiated among populations, $0.063 < F_{ST} < 0.177$, $P < 0.001$. Significant

**Table 2 Microsatellite diversity estimates for *Uropohora cardui*, and allele size range for *U. stylata*.** The three *U. cardui* populations Vildbjerg, Frøslev and Neumünster (NMS) are located north, within and south of a genetic transition zone, respectively. $F_{IS}$, inbreeding index; $H_e$, expected heterozygosity; $N_a$, number of alleles. $N_a$ is shown as a grand total and per population (in brackets). The size range estimates for *U. stylata* were based on the combined results from 15 individuals sampled in Denmark and Germany (see 'Materials and Method'). Estimates of deviations from $F_{IS}$ were calculated with Genepop on the web (*Raymond & Rousset, 1995*; *Rousset, 2008*) and Bonferroni corrected for multiple tests.

| Locus | *U. cardui* (N = 59) | | | | | | | *U. stylata* (N = 15) | |
| | Vildbjerg N = 20 | | Frøslev N = 20 | | NMS N = 19 | | $N_a$ | Size range | $N_a$ |
| | $H_e$ | $F_{IS}$ | $H_e$ | $F_{IS}$ | $H_e$ | $F_{IS}$ | | | |
|---|---|---|---|---|---|---|---|---|---|
| Uc01 | 0.747 | −0.004 | 0.859 | 0.127 | 0.896 | 0.300** | 17 (7/10/12) | – | |
| Uc02 | 0.707 | 0.009 | 0.697 | 0.140 | 0.561 | −0.031 | 5 (4/5/3) | 262–290[b] | 3 |
| Uc03 | 0.786 | 0.045 | 0.764 | 0.281 | 0.746 | 0.012 | 10 (8/6/4) | 267–285[b] | 4 |
| Uc04 | 0.900 | 0.056 | 0.867 | −0.038 | 0.873 | −0.085 | 21 (13/11/10) | 318–478 | 21 |
| Uc05 | 0.596 | 0.161 | 0.797 | 0.310*** | 0.790 | 0.333[a] | 11 (5/7/7) | –[b] | |
| Uc06 | 0.613 | −0.142 | 0.846 | 0.503*** | 0.763 | 0.172 | 14 (5/9/9) | – | |
| Uc07 | 0.901 | −0.054 | 0.836 | −0.017 | 0.811 | 0.027 | 18 (12/9/8) | 411–435[b] | 3 |
| Uc08 | 0.774 | −0.099 | 0.757 | −0.057 | 0.747 | −0.057 | 11 (5/6/7) | – | |
| Uc09 | 0.857 | 0.124 | 0.811 | −0.049 | 0.779 | −0.013 | 10 (8/7/7) | 276–440 | 11 |
| Uc10 | 0.859 | 0.011 | 0.876 | −0.027 | 0.785 | 0.062 | 13 (8/9/8) | –[c] | |
| Uc11 | 0.837 | 0.044 | 0.701 | 0.216 | 0.781 | 0.124 | 11 (7/6/7) | 206–245 | 7 |
| Uc12 | 0.874 | 0.886*** | 0.900 | 0.389*** | 0.814 | 0.418*** | 13 (7/10/7) | 174–262 | 8 |

**Notes.**
[a] Indication of null allele despite non-significant $F_{IS}$.
[b] Inconsistent amplification.
[c] Loci Uc05 and Uc10 amplified same product.
The *P* values, **$P < 0.01$, ***$P < 0.001$, are the original values.

genetic differentiation was found between all population pairs, $F_{ST(\text{north-transition})} = 0.076$, $F_{ST(\text{north-south})} = 0.122$, $F_{ST(\text{transition-south})} = 0.128$.

Cross-species amplification in *U. stylata* was successful for four loci (Uc04, Uc09, Uc11, Uc12), whereas three loci (Uc02, Uc03, Uc07) amplified inconsistently using the amplification protocol used for *U. cardui*. Fragment sizes of Uc03 partially overlapped with those of Uc09 (both PM1), making an evaluation of the less well amplifying Uc03 difficult. Two loci, Uc05 and Uc10 (both PM2), amplified identical fragment profiles. Future studies of *U. stylata*, or other *Uropohora* spp., should take notice of re-evaluating amplification protocols and primer mix compositions.

## GBS

Neither DNA preparation/extraction methods nor treatments with or without RNase affected GBS library quality and sequencing success. Although treatment with RNase significantly improved the mean ratio of 260/280 by approaching ≈1.80 (no RNase = 2.066, with RNase = 1.864; $t = 9.59$, 16 *df*, $P < 0.001$), the mean number of reads per pool was higher, though not significantly, for DNA without RNase treatment (Appendix S2). PyRAD analysis quantified 645,164–2,640,743 reads and 7,615–10,002 loci per individual. The mean depth of clusters per individual with depths greater than 7 was 31.7–88.1. Assembled GBS reads have NCBI accession numbers SAMN06241878–SAMN06241895. The parameter settings "minimum coverage of individuals" (mincov) = 10 and 14 and

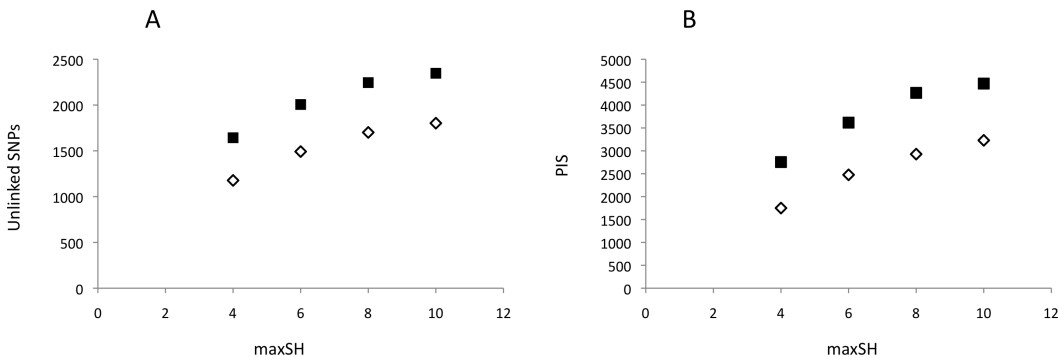

**Figure 2  Genetic variability of recovered GBS loci as a function of pyRAD parameter "maximum number of individuals with heterozygotic sites" (maxSH).** (A) Number of unlinked SNPs. (B) Number of parsimony informative sites, PIS. Genetic variability was estimated for 10 (black squares) and 14 (open diamonds) "minimum coverage of individuals" (mincov).

"maximum number of heterozygote samples" (maxSH) = 4, 6, 8 and 10 identified between 1,177 (14 mincov–4 maxSH) and 2,347 (10 mincov–10 maxSH) unlinked SNPs and 1,750 and 4,469 parsimony informative sites, respectively (Fig. 2). Regression analysis showed that the frequency of polymorphic sites, 0.0034–0.0040, did not increase significantly with the number of reads ($t = -0.61$, $df = 16$, $P = 0.55$) whereas expected heterozygosity did ($t = 2.46$, $df = 16$, $P = 0.03$). The latter was caused by a significant positive relationship between the number of reads and the number of loci recovered for analysis, $t = 3.59$, $df = 16$, $P > 0.01$. *Post hoc* analysis showed that this result was influenced by the three individuals with the lowest number of recovered loci (ca. 200–300 loci less than the other individuals) and implied that a plateau was reached above *c.* 1,000,000 reads using in our protocol. The summary results are presented in Appendix S3.

Structure analyses were performed on loci recovered with the parameter settings 14–6 (i.e., mincov = 14, maxSH = 6), 14–10, 10–6 and 10–10. The parameter set 14–6 recovered 1,492 unlinked SNPs (the second lowest number of SNPs found). For 14–6, the methods of *Evanno, Regnaut & Goudet (2005)* and *Falush, Stephens & Pritchard (2003)* estimated $K = 2$ (Fig. 3A) and $K = 4$ (Fig. 3B) whereas membership proportions identified three clusters corresponding to the three populations (Fig. 3C), thus repeating the observed pattern found for microsatellites without inclusion of population prior. Increasing the number of loci (=unlinked SNPs) by reducing the number of recovered individuals/locus (e.g., 10–6) and/or allowing more heterozygotic sites across samples (e.g., 14–10) resulted in an identical number of optimal clusters, $K = 3$, identified by the three clustering methods (Figs. 3D, 3E, 3G, 3H, 3J and 3K). Genetic differentiation among populations (AMOVA) based on parameter set 14-6 (1,492 unlinked SNPs) included 508 loci with less than 0.05 missing data: $F_{ST} = 0.197$, pairwise estimates: $F_{ST(north-transition)} = 0.165$, $F_{ST(north-south)} = 0.208$, $F_{ST(transition-south)} = 0.220$, all $P < 0.001$. Locus-wise AMOVA based on 701 polymorphic loci found 99 loci (14.1%) with significant differentiation, $0.20 < F_{ST} < 0.91$, $P < 0.05$. Expected heterozygosity/population across 1,492 loci was: North = 0.317, transition zone = 0.331, South = 0.327.

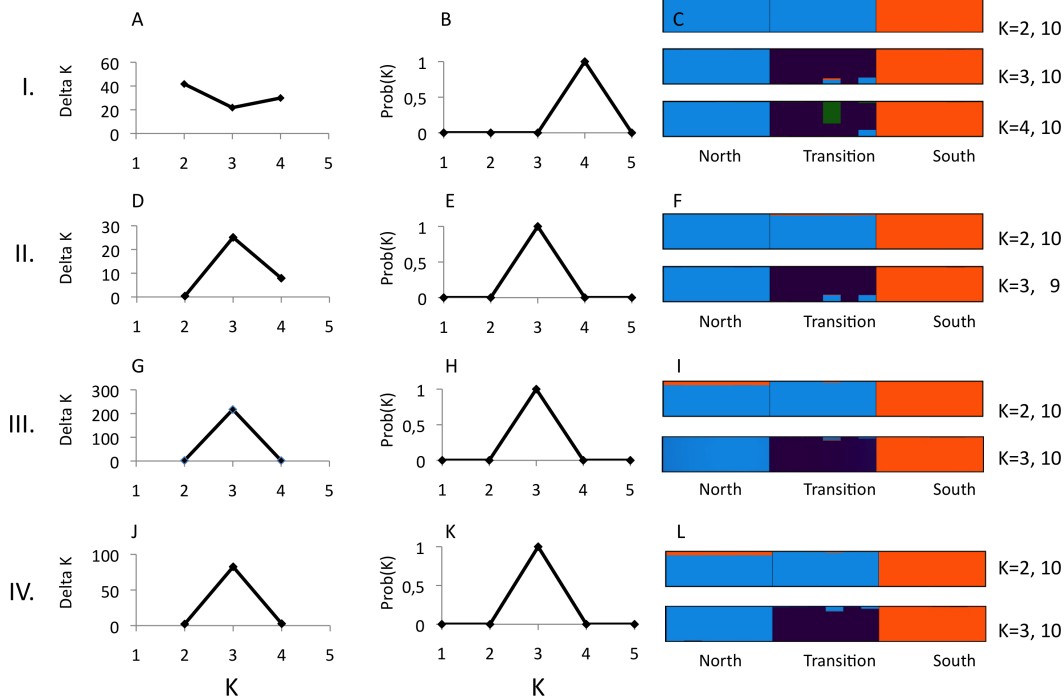

**Figure 3  Structure analysis of GBS data for three parapatric populations (North $N = 6$, Transition $N = 6$, South $N = 6$) relative to combinations of pyRAD parameters "minimum coverage of individuals" (mincov) and "maximum number of individuals with heterozygotic sites" (maxSH).** Row I.: mincov = 14, maxSH = 6 (1,492 unlinked SNPs). Row II.: mincov = 14, maxSH = 10 (1,801 unlinked SNPs). Row III. : mincov = 10, maxSH = 6 (2,006 unlinked SNPs). Row IV.: mincov = 10, maxSH = 10 (2,362 unlinked SNPs). The figure shows the optimal number of $K$ clusters identified by delta $K$ (A, D, G, J) (*Evanno, Regnaut & Goudet, 2005*), the probability of $K$ (B, E, H, K) (*Falush, Stephens & Pritchard, 2003*), and individual membership probabilities (visualised by different colours) (*Pritchard, Stephens & Donnelly, 2000*) for best $K$'s of each parameter set (C, F, I, L) based on the admixture model. Numbers after $K$ (e.g., 10) represent the number the partitions found in 10 runs. Calculations were done with Clumpak (*Kopelman et al., 2015*).

## DISCUSSION

The microsatellite and GBS markers characterised in this study identified identical divisions among three parapatric *U. cardui* populations, thus providing evidence for genetic separation of these populations since the first observation of genetic divergence in 2001 (*Steinmetz, Johannesen & Seitz, 2004*). Many and highly differentiated loci in both marker systems suggest discrete genetic differences among the studied populations. Although our Miseq data did not permit making a high-resolution genome assembly of *U. cardui* for mapping the distribution of SNP diversification across the genome, *in silco* digestion of scaffolds of the tephritids *C. capitata* and *B. tryoni* with EcoR1 found similar fragment length distributions among both scaffolds and species. Hence, we predict genetic divergence among the parapatric *U. cardui* populations will be present in independent genome regions. Indeed, genomic studies of parapatric/sympatric populations/species have found genome-wide divergence patterns (*Renuat et al., 2013*; *Michel et al., 2010*; *Feulner*

*et al., 2015*; *Marques et al., 2016*), which conflicted with theoretical expectations of genomic islands of divergence for local environmental adaptation in the presence of gene flow.

The two marker systems provided different locus-specific levels of genetic diversity that alone or in combination may give insights into such diverse research fields as paternity testing, hybrid status and phylogeograhy. Although the marker systems found identical population-divisions they also differed in estimates of individual affiliations, and GBS was partly sensitive to paralog filtering assumptions. The differences are partly due to different levels of locus-specific heterozygosity, where highly multi-allelic microsatellite loci increase the probability of detecting close relatives but lower the likelihood of assigning "unrelated individuals" to a population when the number of individuals is low relative to the level of polymorphism per locus. The lower estimate of mean genetic differentiation for microsatellites, $F_{ST} = 0.109$, being half of that observed for GBS, $F_{ST} = 0.197$, is related to higher per-locus variability of the former marker system. Still, $F_{ST} > 0.10$ is considered high for highly polymorphic microsatellites, and similar estimates have been found in host-race studies (e.g., *Kempf et al., 2009*; *Imo, Maixner & Johannesen, 2013*).

For GBS, the probability of population assignment increased with the number of unlinked SNPs, but even the relatively low number recovered with the conservative parameter set 14-6 (Fig. 2) identified individual memberships to three populations with high probabilities (Fig. 3). Considering that the individual GBS coverage in 1/5 lane was mostly >40, increasing the number of individuals per pool without quality loss should be possible. Standard DNA extraction procedures did not influence data quality, given adequate concentrations of DNA.

The likelihood for genome-wide differentiation and the observation of alternative sex linkage at microsatellite locus 12 might indicate that genomic rearrangements and/or sex determination are involved in diversification among the parapatric populations. Sex determination in Diptera can vary considerably (*Vicoso & Bachtrog, 2015*), and in the Tephritidae both male and female heterogamy is known (*Bush, 1966*). In *U. cardui* ($2N = 12$), where the sex-determination system is not known (*Mainx, 1976*), karyotype variation in the number of dot chromosomes has been found in southern Germany ($2N = 13$–$14$) (*Ponisch & Brandl, 1992*). Verifying rearrangements and/or sex linkage with GBS data will require more samples than used here due to the combined action of different sex linkage among populations and variable genome representations among individuals. Future studies of *U. cardui* from the transition area should focus on whether genomic rearrangements and sex determination interact with mating behaviour and influence rates of gene flow.

## ACKNOWLEDGEMENTS

We thank Christiane Stürzbecher and Dagmar Klebsch for helping establish the multiplex PCR and GBS procedures. High performance computing for GBS analyses was done on the supercomputer Mogon, including advisory services offered by Johannes Gutenberg University Mainz (hpc.uni-mainz.de), which is a member of the AHRP and the Gauss Alliance e.V.

### Funding

This work was supported by the Deutsche Forschungsgemeinschaft (grant JO-325/5-1). The funders had no role in study design, data collection and analysis, decision to publish, or preparation of the manuscript.

### Grant Disclosures

The following grant information was disclosed by the authors:
Deutsche Forschungsgemeinschaft: JO-325/5-1.

### Competing Interests

Betina Ebner and Sven-Ernö Bikar are employees of StarSEQ GmbH.

### Author Contributions

- Jes Johannesen conceived and designed the experiments, performed the experiments, analyzed the data, contributed reagents/materials/analysis tools, wrote the paper, prepared figures and/or tables, reviewed drafts of the paper, sampled and reared specimens.
- Armin G. Fabritzek performed the experiments, analyzed the data, reviewed drafts of the paper, prepared GBS pipelines.
- Bettina Ebner performed the experiments, reviewed drafts of the paper, assembled Miseq libraries for microsatellite identification.
- Sven-Ernö Bikar conceived and designed the experiments, performed the experiments, contributed reagents/materials/analysis tools, reviewed drafts of the paper, assembled Miseq libraries for microsatellite identification.

### DNA Deposition

The following information was supplied regarding the deposition of DNA sequences:
The sequences described here are accessible via GenBank accession numbers KT923909–KT923920 (microsatellite loci) and SAMN06241878–SAMN06241895 (assembled GBS loci).

### Data Availability

Dryad Digital Repository: http://dx.doi.org/10.5061/dryad.j1828.

### Supplemental Information

Supplemental information for this article can be found online at http://dx.doi.org/10.7717/peerj.3582#supplemental-information.

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
