# Peer review of "Characterisation of microsatellite and SNP markers from Miseq and genotyping-by-sequencing data among parapatric Urophora cardui (Tephritidae) populations"

_PeerJ, doi:10.7717/peerj.3582_

## Round 0.1 · original submission · Minor Revisions

· Academic Editor

Minor Revisions

Thank you for your submission, both reviewers suggest minor changes and I agree with them. Please make the corrections and resubmit.

Laura

·

Basic reporting

The article under review by Johannesen and co-authors is clear, well-written and of interest to all population geneticists and genomists. Comparisons of microsatellite and GBS markers differentiation patterns are important and every new case is informative – this one is clean and rigorous in both molecular and the analytic parts. The article should be accepted with minor revisions and added information as detailed below.
Tables & Figures
Table 2. In the caption, please indicate the test used to estimate significance of HWE (esp. if Bonferroni correction was used), even if it is written in the main text.
Figure 2 & 3. Captions should remind the reader of the number of individuals analyzed in the structure runs for each pop. Axis legends of the DeltaK graphs are too small: the font could easily be increased 2-fold. Graph titles in Fig 3 should be indicated only once, as was done for Fig. 2. For figure 3. Please keep the same color schemes between Structure bar graphs (e.g. north is light blue in 1 and 2, but dark blue in 3 and 4).

Experimental design

Introduction
A paragraph should be added to justify the utility of comparing microsatellite and GBS data. Why use two types of nuclear markers? Are there other studies that point to a meaningful comparison of microsatellite and GBS?
Materials & Methods
Line 80. Please justify why only tetra, penta and hexanucleotides were used – which accumulate within-repeat mutations faster than dinucleotides?
Line 90. What brand and model of sequencer did you use for electrophoresis?
Line 109. Replace Thee by The
Line 141. Please provide the reference for the pyRAD program.
Line 143. Please provide the reference for Structure.
Line 144. Please justify the low number of STRUCTURE iterations.

Validity of the findings

Results
Line 152. The “standard” and “mutant” vocabulary is not useful here, esp. as standard only means the allele found in the individual that was sequenced using NGS. Consider deleting the sentence “mutant alleles…base variation” as this is self-evident for microsatellite markers.
Line 154. “… identical PCR conditions it will be” should be replaced by “it was”.
Line 165. “There was no evidence for stuttering or large allele dropout at any locus”. What about null alleles? How do you explain very high significant Fis values at UC01, UC05 and UC06, as well as high but non-significant values in these and other loci? Is the -0.142 value for Fis at Vildbjerg for Uc06 correct?
Line 209. “were 2 and 4, respectively (replicating the observed pattern for microsatellites)”. K was 2 or 3 in microsatellites, never 4. Thus please remove the text between brackets.
Discussion
Lines 240-243. The sentence is interesting but is there any reference for this affirmation?
Line 244. Comparing Fst values is not valid when not taking heterozygosities into account. It is expected that Fst should level down with increased heterozygosities. Microsatellites are therefore expected to show lower Fst values than GBS (Hedrick PW (1999) Evolution, 53, 313 –318, and one example among many others Hoffman, E.A. et al., 2005. Molecular Ecology, 14(5), pp.1367–1375). If the authors choose to compare, they should compare heterozygosities in a manner similar to the cited papers. They could also drop the implied comparison.
Lines 255-256. Could variability in genome representation among individuals be reduced by improving or changing molecular or analytical methods?

Reviewer 2 ·

Basic reporting

The MS is well written and will just require some editing to correct the typographical errors. It will also be useful to share the MS with a native English speaker to correct the grammar.

Experimental design

The authors have done well in the experimental design. Though the sample size was very low for conclusive analyses of microsatellite data, the authors confirmed this analyses with a second set of markers.

Validity of the findings

The finding are in accordance with literature citation and conform with the insect behavioural and phenotypic trends.

Comments for the author

It will be useful for the authors to include some other tools to do their analyses for instance the population assignments and exclusion for them to confirm the evolutionary trends of the insect in terms of gene flow between the presumed genetic clusters.

Lastly, the authors should add to the discussion a conclusive remark that besides use of molecular markers to resolve the population structure of Urophora cardui, that integrative taxonomic approaches should be undertaken through the use of chemical ecology, behavioural studies, mating compatibility, etc.

---

## Round 0.2 · Minor Revisions

· Academic Editor

Minor Revisions

Please develop the introduction a bit more. The problem statement is not developed enough- what is the damage caused by Cirsium arvense? What strategies have been tried before? In addition, the discussion needs further development.

---

## Round 0.3 · accepted · Accept

· Academic Editor

Accept

Thank you for expanding on the introduction and discussion. It is ready to go now!